# Multitask Multimodal Prompted Training for Interactive Embodied Task Completion

**Georgios Pantazopoulos**[1] **Malvina Nikandrou**[1] **Amit Parekh**[1] **Bhathiya Hemanthage**[1]
**Arash Eshghi**[1,2] **Ioannis Konstas**[1,2] **Verena Rieser**[1,2*] **Oliver Lemon**[1,2] **Alessandro Suglia**[1]

[1]Heriot-Watt University; [2]Alana AI

{gmp2000, mn2002, amit.parekh, hsb2000, a.eshghi,
i.konstas, v.t.rieser, o.lemon, a.suglia} @hw.ac.uk

## Abstract

Interactive and embodied tasks pose at least two fundamental challenges to existing Vision & Language (VL) models, including 1) grounding language in trajectories of actions and observations, and 2) referential disambiguation. To tackle these challenges, we propose an Embodied MultiModal Agent (EMMA): a unified encoder-decoder model that reasons over images and trajectories, and casts action prediction as multimodal text generation. By unifying all tasks as text generation, EMMA learns a *language of actions* which facilitates transfer across tasks. Different to previous modular approaches with independently trained components, we use a single multitask model where each task contributes to goal completion. EMMA performs on par with similar models on several VL benchmarks and sets a new state-of-the-art performance (36.81% success rate) on the Dialog-guided Task Completion (DTC), a benchmark to evaluate dialog-guided agents in the Alexa Arena (Gao et al., 2023).[1]

## 1 Introduction

Embodied AI aims to develop agents that interact with the environment, reason over natural language instructions and visual observations, and plan future actions. Recently, Vision & Language pretraining (VLP) has established new standards across image-based tasks (Bao et al., 2022; Wang et al., 2022a,b; Yang et al., 2021; Zhang et al., 2021) by aligning visual and textual input to generate image captions, answer visual questions, and reason over images. As a result, VLP promotes learning of important skills transferable to embodied tasks.

Interactive and embodied tasks present significant challenges for VLP models including 1) grounding language in environments where the interpretation of instructions depends not just on static images, but on trajectories of actions and observations, and 2) referential disambiguation, where agents must interact with users — often as clarifications — to successfully disambiguate referents. In contrast to language-only planners (Huang et al., 2022a; Min et al., 2021), which cannot generate feasible plans without prompts that encode all the visual nuances — colors, attributes, relational information, inter alia — as text; instead VLP models can ground language directly to images. However, to enable grounded decision-making, VLP models must generalize from static images to trajectories that express how a situation evolves over time (McClelland et al., 2020). Additionally, in language-guided collaborative tasks, referents for target objects may be ambiguous, depending on the complexity of the visual scene. Therefore, clarifications must be adopted to resolve uncertainty between referents and act upon the correct target object (Madureira and Schlangen, 2023).

To address both of these challenges, we develop EMMA: a unified approach which casts several VL and embodied tasks, such as image captioning and action execution, as text generation inspired by Cho et al. (2021). By doing so, EMMA is able to learn a *language of actions* which facilitates transfer across tasks. Concretely, EMMA is a multitask encoder-decoder model, which encodes language and pairs of object-centric representations with sentinel tokens. As a result, EMMA can encode both trajectories and clarifications to reference individual frames and objects across an entire trajectory.

We focus on dialog-enabled task completion, which requires an agent to follow natural language instructions, perform navigation and object manipulation actions, and utilize dialog interactions to disambiguate visual observations. Similar to previous modular approaches (Min et al., 2021), we break down the embodied task into separate processes for search and action execution. The search process combines exploration with visual grounding. Differently to prior work that only uses object

---

*Now at Google DeepMind

[1]Code available on this link

labels from object detectors directly (Min et al., 2021; Pashevich et al., 2021), EMMA uses both language and visual representations to discriminate between objects of the same class (e.g., discerning between a 'red' and a 'green' desk). Separately, the action execution process predicts the navigation and object interaction actions as textual tokens. Notably, a key distinction in our approach is that we perform all tasks using a single VL model.

We evaluate our approach on the Dialog-guided Task Completion (DTC) benchmark (Gao et al., 2023). Our experiments show the benefits of the multitask VLP, and the importance of data augmentations for completing multi-step instructions. By exploiting clarifications, EMMA improves the success rate by 3.55%. Finally, when fine-tuned on DTC, EMMA can transfer the downstream action execution task back to real-world images, showcasing how using text tokens to represent actions in the environment enables cross-domain generalization.

## 2 Related Work

**Vision & Language Pretraining**   Early methods for developing VLP models rely on multimodal encoders with object-centric or patch representations (Chen et al., 2020; Kim et al., 2021; Li et al., 2021, 2020; Lu et al., 2019; Singh et al., 2022a; Tan and Bansal, 2019). While effective, these methods introduce task-specific heads for each downstream task. Generative and unified architectures represent an alternative approach for tackling VL tasks with a single model. Previous work on unified models explores region proposals (Cho et al., 2021; Gupta et al., 2022), or patch features (Wang et al., 2022a,b; Yang et al., 2022). More recently, the paradigm of connecting vision-only and language-only experts allows leveraging unimodal pretraining for generative and multimodal models (Alayrac et al., 2022; Driess et al., 2023; Li et al., 2023; Liu et al., 2023; Tsimpoukelli et al., 2021; Zhu et al., 2023). By casting every task as a text generation task, unified VLP models are transitioning from specialized to generalist models. Therefore, adopting a similar multitask pretraining approach enables EMMA to learn a language that describes actions conditioned on visual observations.

**Embodied AI Approaches**   Previous approaches on interactive task completion (Shridhar et al., 2020) involve both end-to-end (Pashevich et al., 2021; Suglia et al., 2021) and modular architectures (Blukis et al., 2022; Kim and Lee, 2023; Liu

et al., 2022; Min et al., 2021; Singh et al., 2021). State-of-the-art works employ a modular architecture that incrementally builds a semantic map of the environment used by symbolic planners. However, these methods often make strong domain-specific assumptions, or they are typically trained from scratch, underutilizing recent progress in VLP.

An exciting direction is leveraging large-scale pretrained models. Recent works (Ahn et al., 2022; Driess et al., 2023; Huang et al., 2022a; Singh et al., 2022b; Zeng et al., 2022) use a large language model or a mixture of pretrained models for object manipulation. Additionally, Huang et al. (2022b); Shridhar et al. (2021) use text to describe the environment state and employ a language model for planning. Different to previous work, we use a VLP model to ground instructions on visual scenes. For example, given the instruction *'Get the milk from the fridge'*, EMMA executes the correct action sequence by inferring whether the fridge door is open or not from its visual observations.

Dialog-enabled interactive task completion introduces an additional layer of complexity as the agent often needs to incorporate clarification context before acting on the environment (Gao et al., 2023, 2022; Padmakumar et al., 2022). Both end-to-end and modular architectures underperform in these tasks, which Min et al. (2022) argues is due this is due imitation learning using few and sub-optimal demonstrations. EMMA is trained using expert demonstrations and interactions in the form of question-answer pairs about the visual scene. These clarifications do not involve complex dialog phenomena (e.g., repair), yet are sufficiently informative to disambiguate target objects.

## 3 Task Description

The DTC benchmark evaluates dialog-enabled agents to complete missions in the Alexa Arena, a simulated environment with multi-room layouts (Gao et al., 2023). Each example corresponds to a mission completed by an expert planner and annotated by three humans. Each instruction is optionally accompanied by a clarification question-answer pair. Navigation is performed with primitive actions, but also with the GoTo action, which allows moving to different rooms, or objects. Along with the GoTo action, the agent needs to predict the name of the room or an object mask within the current observation. Each room also contains a set of viewpoints that differ between layouts and can

Figure 1: Overview of the EMMA architecture. EMMA encodes vision and language inputs with modality-specific layers before providing the input to the bidirectional multimodal encoder. Scene embeddings act as global features for a frame. Object embeddings correspond to a combination of region features, object coordinates, and their identities within the frame sequence. Language input is concatenated with **task-specific prompts**. The shared autoregressive language decoder treats every task as a text generation task.

be visited by the agent to explore the room. To perform an action on an object, the agent needs to provide the action type and the mask of the object. The set of supported actions for an object is determined by its affordances such as openable, toggleable, inter alia (see Appendix C for details).

## 4 EMMA

EMMA is an encoder-decoder model following the architecture of BART-base (Lewis et al., 2020). As shown in Figure 1, both vision and language inputs are embedded through modality-specific projection layers, concatenated into a sequence of embeddings, and fed into a single-stream encoder. EMMA uses a shared decoder across all pretraining tasks with sentinel tokens in the vocabulary to allow referencing specific image frames and regions.

**Text Embedding** For the language input, we apply sub-word byte-pair encoding (Sennrich et al., 2016) with a vocabulary of 10K tokens extracted from our pretraining data. Each token is represented by the sum of its word and absolute positional embedding (Lewis et al., 2020). Similar to Sanh et al. (2022), we use natural language prompts as task prefixes to prompt the model for each task.

**Scene and Object Embeddings** EMMA is capable of encoding sequences of images. For each frame of the visual input, we extract global scene features representing the entire frame and a maximum of $n = 36$ region features. To reference a frame or an object within it, the language vocabulary is extended with sentinel tokens in the form of <frame_token_i> and <visual_token_j>. After projecting the visual features to the encoder's dimensionality, we add spatial, temporal, and visual

sentinel token embeddings. Spatial embeddings encode the 2D position of the region within the frame by its normalized bounding box coordinates, while temporal embeddings encode the temporal order within the sequence using a frame sentinel token.

The choice between object-centric and patch representations for embodied tasks remains an open question. Although patch representations contain explicit information about the semantic segmentation of an image (Caron et al., 2021), their static grid structure has not been proven beneficial in previous embodied tasks (Driess et al., 2023; Jiang et al., 2023). Furthermore, patch representations lead to an increase in both input length, as more visual tokens are required to represent a frame, and output length, as the model must generate coordinate tokens instead of a single sentinel token to reference an object. For these reasons, we opted for object-centric representations.

### 4.1 Pretraining Setup

We formulate seven VL tasks in a text-to-text framework motivated by established pretraining tasks (Cho et al., 2021; Wang et al., 2022b). Pretraining details are provided in Appendix A. Below we give a brief description of each task:

**Masked Language Modeling:** Given an image description, we mask each word with probability 0.3. The model must learn to reconstruct the original input by predicting the masked words.

**Image-Text Matching:** Determine if a pair of visual-text inputs match by predicting true/false after we randomly combine the visual input with either the correct caption or a sampled caption.

**Visual Question Answering:** Answer a question about the input image.

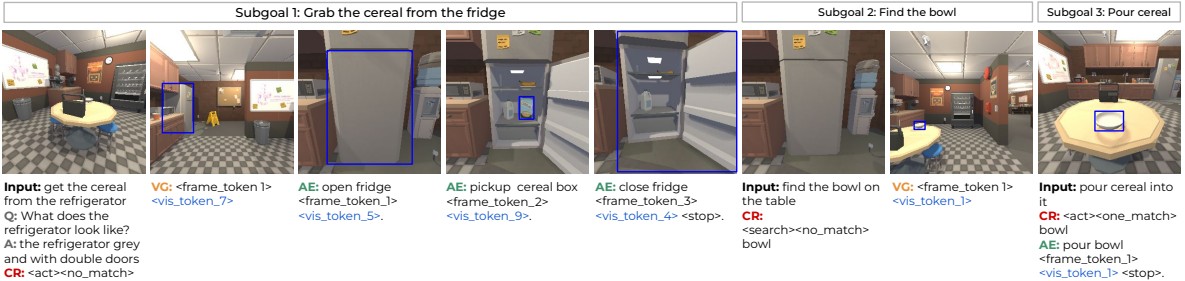

Figure 2: Example trajectory where the objective is to pour cereal into the bowl. The user instructions and our agent response are shown on the left and right of each frame respectively. At each timestep the agent is provided with the current view, a user instruction (**Input**), and optionally a clarification question (**Q**) and answer (**A**). The Contextual Routing task (**CR**) determines whether the agent is going to use the Action Execution task (**AE**) to interact with the environment, or the Visual Grounding task (**VG**) to search for an object in the scene.

**(Dense) Captioning:** Produce a textual description of the overall image or a specified region denoted by a visual sentinel token.

**Visual Grounding:** Given a description of an image region, the model must predict the visual token for the region that matches this description.

**Relationship Prediction:** Describe the relationship between two regions of an image. The output follows the format: *Subject Attributes, Subject, Relationship Predicate, Object Attributes, Object*.

## 4.2 Interactive Task Completion

Our agent uses three distinct yet complementary tasks to complete the DTC missions: Contextual Routing (CR), Action Execution (AE), and Visual Grounding (VG). As shown by Figure 2, we use a single model to tackle all three tasks. Since CR & AE are downstream-specific tasks, we introduce new prompts when fine-tuning.

**Contextual Routing (CR)** Given the current instruction and observation, the CR task determines whether the agent should act or search for an object, disentangling the decision of what to do next from how to achieve it. The output of the model follows a structured hierarchical scheme (Figure 3). The first token determines whether the instruction refers to an action or to search for an object. The second token indicates if there are no objects present, one object present, or multiple matches of the object mentioned in the instruction. Finally, for non-navigation actions, the model generates the name of the target object.

**Action Execution (AE)** The agent acts on the environment if the output of the CR task is `<act><one match>` or `<act><multiple`

`matches>`. The input to the model consists of the current frame, the instruction, and the clarification pair. We use the `<follower>` and `<commander>` tokens as prefixes to each dialog turn. The model predicts the action type, the name of the object, and, for manipulation actions, the frame and visual token id that correspond to the object (Figure 1). Each action yields a new observation concatenated with the previous visual context to make a new prediction. The period character ('.') delimits actions, and the `<stop>` token marks the end of the trajectory for that instruction.

**Visual Grounding (VG)** For `<search>` and `<act><no match>` predictions, the agent tries to find the object outputted by the CR task. We iterate over viewpoints in a room to collect panoramic views and use the pretraining visual grounding prefix with the instruction as input to the model (Appendix D). If the instruction matches with a visual token, the model outputs that token; else, the model outputs 'no OBJECT'. For `<act><no match>`, once the object has been found, we invoke the AE task.

## 5 Data Augmentations

Since the trajectories have been generated by an expert planner, the predictions `<search>` and `<act><no match>` used by the CR task are underrepresented in the training data. Training for VG is unsupported as DTC instances only contain the ground-truth object mask for each action but not for all other objects in a scene. Furthermore, preliminary experiments revealed that, when trained solely on instances from the DTC benchmark, our agent struggled to learn associations between frame and visual tokens. For example, when interacting with a fridge across multiple frames, the agent sometimes

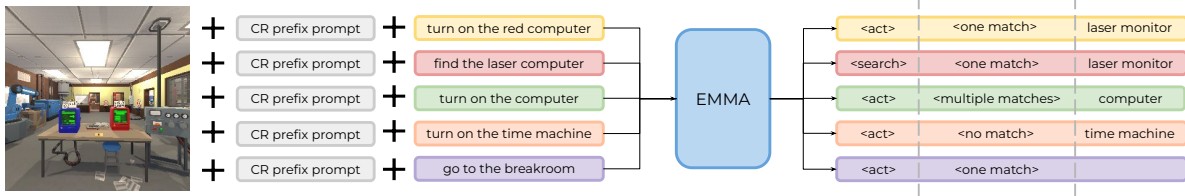

Figure 3: Example outputs for the Contextual Routing (CR) task. The first token determines whether the agent will search or act on the environment. The second token determines if the instruction matches an entity in the scene. Finally, for instructions that require manipulation of objects, the model generates the name of the object.

predicted visual tokens that matched the fridge in a previous rather than the current frame. Finally, there is a mismatch between how the trajectories were annotated and the data the agent is being exposed to. While annotators observed interpolated trajectories, the agent only perceives the state before and after an action. This discrepancy adds significant language variability and complexity, posing challenges to the agent's comprehension.

To address these challenges, we use the vision dataset provided by Gao et al. (2023) to create synthetic instruction data. Additionally, we deploy an early version of our agent to collect mini-episodes of successful trajectories within the Alexa Arena. Supplementary material regarding the data generation process is provided in Appendix E.

**Visual Augmentations** Starting from the images within the vision dataset, we create an additional 180k for training and 88k instances for validation, where each instance is an (image, instruction, action) triplet. When generating target actions, we used the ground truth positions of the agent and the object to ensure the agent was in proximity. For synthetic instructions, we used ChatGPT (OpenAI, 2022) as a starting point to create templates. To select diverse visual scenes, we used CLIP (Radford et al., 2021) to embed the images. We then applied k-means clustering, where the number of clusters is determined by the maximum number of action-object instances. Finally, we selected the image that is closest to each centroid.

**CDF Augmentations** While the visual augmentations are useful for single-step interactions, they cannot help the model on longer trajectories requiring consecutive interaction with the environment. To address this challenge, we use one of the early variants of EMMA to collect more training trajectories. For each mission, we provide the agent with step-by-step instructions and only include successful trajectories in our training data. We refer to

these trajectories as CDF augmentations following the challenge definition format of the Alexa Arena (Gao et al., 2023). Overall, we sampled 38k missions for training and 15k for validation. These missions include: 1) picking up and placing objects from receptacles or containers; 2) picking up objects under the presence of objects acting as distractors (i.e., the same object but with a different color, or the same object placed in a different position); 3) cleaning and filling objects in the sink; and 4) interacting with domain-specific objects (e.g., the color changer and the time machine).

## 6 Experiments & Results

### 6.1 VLP Evaluation

We evaluate EMMA on four image-based downstream tasks including image captioning, visual question answering, referring expression comprehension, and natural language for visual reasoning. We treat all tasks as language generation and fine-tune EMMA separately on each downstream task without introducing any task-specific parameters.

To evaluate the ability of our model to generate image captions, we use the MS-COCO dataset (Lin et al., 2014) and report evaluation results on the Karpathy test split (Karpathy and Fei-Fei, 2015) for BLEU-4 (Papineni et al., 2002), METEOR (Lavie and Agarwal, 2007), CIDEr (Vedantam et al., 2015), and SPICE (Anderson et al., 2016). For visual question answering, we report the VQA accuracy (Antol et al., 2015) on the test-std set of the VQA-v2 dataset (Goyal et al., 2017).

We also evaluate referring expressions comprehension on RefCOCOg (Mao et al., 2016) — a visual grounding task that requires selecting the region described by a given phrase. We use the visual grounding prompts from our pretraining and ask the model to generate the visual token id that matches the given phrase in the image. A predicted region is correct if the intersection over union with

| | # Pretrain Samples | # Params | BLEU-4 | COCO Captioning | | | VQA-v2 Accuracy | RefCOCOg Accuracy@0.5 | NLVR$^2$ Accuracy |
| | | | | METEOR | CIDEr | SPICE | | | |
|---|---|---|---|---|---|---|---|---|---|
| VL-T5 (Cho et al., 2021) | 7.6M | 172M | 34.5 | 28.7 | 116.5 | 21.9 | 70.3 | 71.3 | **73.6** |
| VL-BART (Cho et al., 2021) | 7.6M | 172M | 25.1 | 28.7 | 116.6 | 21.5 | 71.3 | 22.4 | 70.3 |
| UniTAB (Yang et al., 2022) | 8.1M | 211M | 36.1 | 28.6 | 119.8 | 21.7 | 71.0 | **84.5** | — |
| OFA-base (Wang et al., 2022b) | 21.3M | 182M | 41.0 | 30.9 | 138.2 | 24.2 | 78.1 | 82.3 | — |
| EMMA | 10.2M | 133M | **36.5** | **29.7** | **122.3** | **22.5** | **73.2** | 80.3 | 70.3 |

Table 1: Performance of the pretrained model on downstream image-based tasks. We report the number of pretraining samples as the number of image-text pairs. OFA uses additional vision-only and language-only data.

the ground truth region is larger than 0.5.

In the NLVR$^2$ dataset (Suhr et al., 2019), given a caption and two images, the model must predict whether the caption matches both images. We provide the two images directly to the model and prefix the caption with the image-text matching prompt. The model then predicts either true or false.

Table 1 illustrates the results of our model. We compare EMMA against similar VL models — i.e., single-stream encoder-decoder models like VL-T5 and VL-BART (Cho et al., 2021), UniTab (Yang et al., 2022), and OFA (Wang et al., 2022b). Among these models OFA-base achieves state-of-the-art performance, however, it has $1.37\times$ more parameters than EMMA and is pretrained with nearly double the amount of data. On RefCOCOg, our model achieves competitive performance with UniTAB, even though it is almost half the size. Cho et al. (2021) hypothesize that the reason for the poor performance of VL-BART on RefCOCOg is that the use of absolute positional embeddings leads to memorization during training. However, EMMA achieves competitive performance despite using absolute positional embeddings. We believe this is due to our sampling procedure that takes into account the dataset imbalance in pretraining (see Appendix A for details).

## 6.2 Interactive Task Completion

We compare two variants of EMMA: a modular agent with two experts that are each fine-tuned on the downstream-specific tasks (i.e., CR & AE), and a unified model fine-tuned with all the tasks. As shown in Table 2, both agent variants outperform the baseline models. Additionally, the modular agent performs on par with the best-performing models on the leaderboard, while the unified agent slightly outperforms the state-of-the-art. We believe that the synergy between tasks provides an advantage to the unified model.

| | MSR (↑) | NRA (↓) | QA |
|---|---|---|---|
| *Leaderboard:* | | | |
| GauchoAI | 36.47 | — | — |
| SEAGULL | 30.98 | — | — |
| Kingfisher | 22.37 | — | — |
| *Baseline:* | | | |
| NS (Gao et al., 2023) | 19.32 | 11.73 | ✗ |
| NS (Gao et al., 2023) | 22.80 | 12.73 | ✓ |
| VL (Gao et al., 2023) | 18.19 | 11.82 | ✗ |
| VL (Gao et al., 2023) | 34.20 | 8.82 | ✓ |
| *EMMA:* | | | |
| EMMA-modular | 33.76 | 8.91 | ✗ |
| EMMA-modular | 33.95 | 9.05 | CR |
| EMMA-modular | 35.16 | 8.92 | ✓ |
| EMMA-unified | 33.26 | 8.79 | ✗ |
| EMMA-unified | 33.59 | 8.89 | CR |
| EMMA-unified | **36.81** | **8.69** | ✓ |

Table 2: Mission Success Rate (MSR) and the Number of Robot Actions (NRA) of EMMA against top-3 leaderboard and baseline models. CR denotes QA based on the CR task (<act><multiple matches>).

We also compare the success rate under three conditions for the inclusion of clarifications: 1) no clarifications, 2) clarifications whenever they are available for an instruction, 3) clarifications whenever they are available, and the CR predicts an ambiguous referent, that is <act><multiple matches> is predicted. We observe that both versions of EMMA improve when clarification exchanges are added to the input. Performance gains are marginal in the CR case, however, the agents benefit substantially when clarifications are always provided. Note that the role of the CR task is not to decide when to clarify but to drive the agent to either act or search the environment. On top of that, during training, the agent learns to rely on all available clarification types.

**Performance per Mission Type** Table 3 shows the performance of the modular and the unified agents in every mission category along with the average number of ground-truth actions required

|  | EMMA-modular | | EMMA-unified | | Average # of ground truth actions |
|---|---|---|---|---|---|
|  | QA ✗ | QA ✓ | QA ✗ | QA ✓ |  |
| breakObject | 32.22 | 35.56 | 33.33 | 41.11 | 8.43 |
| clean&deliver | 9.20 | 13.79 | 18.39 | 20.69 | 12.65 |
| color&deliver | 0.00 | 0.00 | 0.00 | 0.00 | 14.00 |
| fill&deliver | 29.17 | 29.17 | 27.08 | 27.08 | 12.87 |
| freeze&deliver | 12.50 | 16.67 | 20.83 | 20.83 | 15.75 |
| heat&deliver | 5.13 | 7.69 | 10.26 | 10.26 | 16.92 |
| insertInDevice | 37.29 | 38.42 | 39.55 | 45.76 | 7.44 |
| pickup&deliver | 18.95 | 21.75 | 18.25 | 21.40 | 7.57 |
| pourContainer | 18.95 | 47.86 | 41.03 | 46.15 | 8.33 |
| repair&deliver | 11.11 | 20.37 | 12.96 | 22.22 | 17.05 |
| scanObject | 69.37 | 63.06 | 63.96 | 62.16 | 3.91 |
| toggleDevice | 67.62 | 68.57 | 62.86 | 65.71 | 3.94 |

Table 3: MSR of EMMA for each mission category.

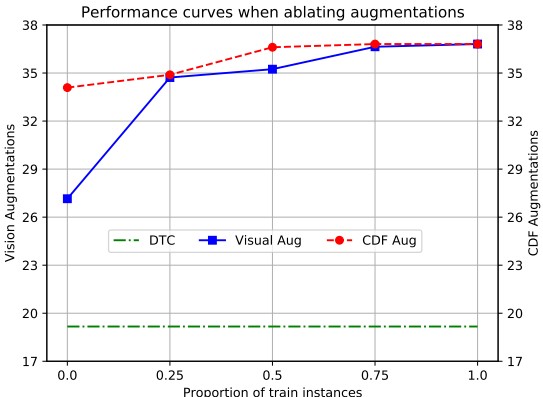

Figure 4: MSR against data ablations. The DTC line is the performance of the model after training exclusively on the DTC data, without any augmentations. Since the visual grounding task is not modeled by the DTC benchmark, we keep the grounding instances and only downsample the action execution instances.

|  | EMMA-modular | | | EMMA-unified | | |
|---|---|---|---|---|---|---|
|  | QA ✗ | QA ✓ | Gain | QA ✗ | QA ✓ | Gain |
| Description | 59.05 | 66.40 | +7.35 | 62.84 | 67.98 | +5.14 |
| Direction | 73.43 | 76.56 | +3.13 | 79.41 | 83.82 | +4.41 |
| Location | 62.27 | 66.45 | +4.18 | 64.27 | 66.66 | +2.39 |
| Reference | 59.05 | 64.09 | +5.04 | 62.02 | 66.56 | +4.54 |
| Other | 75.00 | 75.00 | +0.00 | 79.41 | 79.41 | +0.00 |

Table 4: Object localization accuracy per question type for the modular and unified model. QA ✗ columns refer to the performance of the model on the same instructions but without clarification.

by the expert planner to solve the mission. Both agents perform well on small trajectories, whereas for longer trajectories the unified approach yields better performance. Clarifications play a key role across all mission types with the exception of *scanObject* missions, a behavior which is also shown in baseline models (Gao et al., 2023). Interestingly, both agents are completely unable to solve *color&deliver* missions. Based on the predictions of the model, the main challenge is finding the correct receptacle at the end of the trajectory, which can be easily confused with other similar objects.

**Impact of Clarification Type** Next, we are interested in identifying which clarifications help the model disambiguate object referents. Similarly to Chiyah-Garcia et al. (2022), we compare the object localization performance of the model with and without clarification. Table 4 illustrates the localization accuracy for the modular and the unified model across question types. We use the same taxonomy as Gao et al. (2023) with the addition of 'other', which refers to out-of-scope clarifications (e.g, *'What should I deliver'*, while the agent already holds the mug and is instructed to place it on the desk). Both models benefit the most from description clarifications (e.g., *'What does the cartridge look like?'*, *'It is black with a white label'*). The modular agent benefits more from reference and location clarifications, while the unified agent benefits from reference and direction clarifications. We hypothesize that the transfer between tasks enables the unified agent to perform well on instructions even without the location clarifications, thereby the clarifications provide only small improvements.

**Data Ablations** We also provide ablations in the dataset that showcase the effectiveness of our augmentations by ablating the visual as well as the CDF augmentations. Figure 4 depicts the performance of the unified model for both ablations. We observe that the model benefits from both augmentations. Vision augmentations provide performance gains from very early stages since they enable certain properties of the agent (e.g., `<act><no match>`). On the other hand, CDF augmentations provide a steady boost as they enable the agent to solve missions that require longer trajectories. Note that our pretraining consists of image-caption objectives that enable the model to learn to ground text in singular frames. Learning frame-visual token associations is obtained during fine-tuning only. Future work could explore agents pretrained on tasks that favor learning this ability as well.

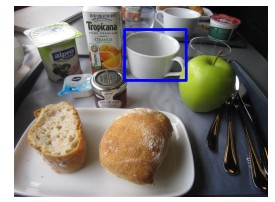 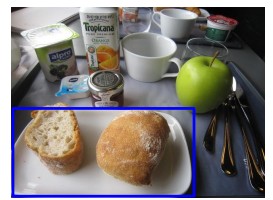 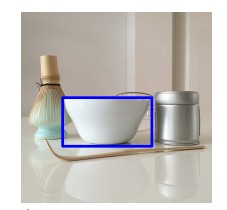 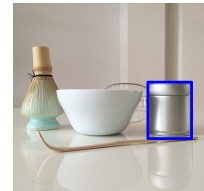

**Instruction:** Put some coffee in the cup. **Output:** pour mug <frame_token_1> <vis_token_17> <stop>.

**Instruction:** Lift the plate with the bread on it. **Output:** pickup plate <frame_token_1> <vis_token_2> <stop>.

**Instruction:** Can you put the spoon into the bowl? **Output:** place bowl <frame_token_1> <vis_token_1> <stop>.

**Instruction:** open up the metallic container. **Output:** open jar <frame_token_1> <vis_token_6> <stop>.

Figure 5: Example transfer of the action prediction task to the visual domain of real images.

## 7 Performance Analysis

**DTC Error Analysis** We inspected the trajectories of our unified model for 120 unsuccessful trajectories (ten per mission type). Overall, the main source of errors stems from the output of the contextual routing component. This task plays a key role in our model since an incorrect <act><one match> prediction triggers the action execution task. In the follow-up timestep, the agent is positioned differently from what is expected and it is forced to act on the environment, which likely leads to an unrecoverable situation. For instance, the agent may be facing an empty desk while being asked to pick up a mug. Consequently, the agent acts in unpredictable ways, resulting in a state from which the agent cannot recover. The incorrect predictions affect mostly <x>&deliver missions, where the agent completed the <x> subgoal, however, failed on the deliver subgoal as it was unable to disambiguate between multiple receptacles.

Furthermore, the output of the action execution task is also susceptible to errors. In long trajectories, we observed a few cases where the model seems to lack temporal understanding. For example, when interacting with objects on consecutive timesteps the model used a visual token to reference the object that matched one of the previous frames in the trajectory. We also noticed a few error cases due to a lack of commonsense reasoning. Some instructions describe properties of an object, like its color and shape without explicit reference to the object but our agent could not make the associations between an object and its properties.

**Visual Domain Transfer** We inspect the model's ability to transfer the action prediction task to real images. We observe qualitatively that after fine-tuning the object detector struggles to detect objects from classes that are not present in the Alexa Arena. However, after reverting back to the base object

| Object Detector | Arena Classes | Non-Arena Classes | Overall |
| --- | --- | --- | --- |
| Fine-tuned | 42.65 | 32.13 | 33.88 |
| Base | 67.85 | 58.85 | 60.35 |

Table 5: Action prediction accuracy for real images from Visual Genome (Krishna et al., 2017).

detector, the EMMA-unified model is able to make reasonable predictions as shown in Figure 5. To quantify the improvement of using the base object detector, we use scene graph annotations of GQA (Hudson and Manning, 2019) by creating synthetic instructions, as described in the Appendix H.

Table 5 shows the accuracy for single-step instructions using real images where the target object can be from any class or the subset of classes that appear in the Alexa Arena. Although EMMA-unified has been fine-tuned on the action prediction task with image inputs from the fine-tuned object detector, we see 26.48% absolute improvement when switching to the base object detector. We leave further exploration of the potential for Sim2Real transfer as future work.

## 8 Conclusion & Future Work

We described EMMA, a unified and multitask model for embodied task completion. EMMA performs on par with VLP models of comparable size in VL benchmarks, and also achieves a new state-of-the-art performance on the DTC benchmark (Gao et al., 2023). We decouple the downstream embodied task into three complementary tasks all solved by the same unified model. Finally, EMMA benefits from interaction with users in the form of clarifications that are crucial for the agent's success, particularly for referential disambiguation.

In terms of future work, our experiments show that trajectory augmentations are required in order to learn frame and visual token associations. However, acquiring additional data might be prohibitive

for target applications where an environment simulator is not available. This requirement could be alleviated by introducing the concept of trajectories in the pretraining. Furthermore, given our visual domain transfer analysis, it would be interesting to explore if visual encoders specialized to various target domains can be plugged into the same multimodal model to allow cross-domain transfer.

## Limitations

We focused on the development of a dialog-enabled agent within an embodied environment. Our design aimed at creating an agent that is as autonomous as possible without relying on external components to support it. We took one step away from modular agents with individual components to a single model that facilitates all tasks. Nevertheless, the search routine is an external component that our agent depends upon. In theory, an agent could learn low-level actions in order to search for an object. However, this is a challenging problem that would require an excessive amount of demonstrations even for simple instructions. Consider the case where the model has to search within multiple containers (e.g., cabinets) for the required object (e.g., a spoon). This would require the agent to learn to summarize the state from the context of history and recognize which cabinets it has already visited before opening the next cabinet. Finally, as shown by our results, our agent does not fully model dialog actions. We used the output of the contextual routing task to request clarifications, however, we did not emphasize on the type of clarifications when making a request. An interesting follow-up work would be to explore how an agent can further incorporate dialog actions similar to the ones supported in the environment and pose clarification requests when necessary.

## Ethics Statement

Embodied AI and physical robots have some well-known ethical trade-offs. On the one hand, they can generate a positive impact, e.g. as part of care settings or helping in disaster zones. On the other hand, they can also lead to negative socio-technical consequences, such as job displacement or dis-empowerment of individuals (Müller, 2021).

Embodied agents based on large-scale VLP inherit additional challenges – some of which they share with large language models, including hallucinations, discrimination and exclusion, malicious uses, and environmental harms (Bender et al., 2021; Dinan et al., 2022; Weidinger et al., 2022).

Furthermore, their multimodal nature creates additional privacy concerns, especially when developing systems for assistive home settings, as is the case in EMMA. In order to address the need for large amounts of in-domain data, EMMA is developed using simulated environments. However, transferring the learned behavior to real-world environments is still an open challenge. For example, VLP models can be prone to distribution shifts, e.g. through naturally occurring noise (Yu et al., 2023), or targeted attacks (Yu and Rieser, 2023). Consequently, embodied agents based on VLP may exhibit unintended behaviors when operating in complex real-world environments. These failures are especially grave when interacting with vulnerable groups, as is the case for care assistants.

It is therefore paramount that these models are released in a responsible way (Bergman et al., 2022), developed, and tested in ecologically valid setups in order to anticipate real-world impact (De Vries et al., 2020). Addressing these issues responsibly and proactively is essential to maximize the benefits while minimizing the potential negative impacts on individuals and society as a whole.

## Acknowledgements

We would like to thank the Alexa Prize team and Amazon Science for their crucial technical and administrative support, along with the resources that greatly contributed towards model development. This work also used the Cirrus UK National Tier-2 HPC Service at EPCC funded by the University of Edinburgh and EPSRC (EP/P020267/1).

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

## A  Pretraining Details

**Vision Encoder**  For encoding images, we use VinVL (Zhang et al., 2021), as it has showed strong performance on established VL benchmarks, keeping it frozen for pretraining and only fine-tuning it on the Alexa Arena.

**Pretraining Dataset**  We pretrain our model on publicly available VL datasets including COCO captions (Lin et al., 2014), Conceptual Captions 3M (Sharma et al., 2018), GQA (Hudson and Manning, 2019), VQA (Goyal et al., 2017), and Visual Genome (Krishna et al., 2017). Table 6 shows the statistics for our compiled pretraining corpus. Overall we used 10.4M examples on 3.3M different images. We used custom task-specific prefixes for each task. Table 7 illustrates example input-output pairs for each pretraining task, while Table 8 illustrates the prompts used for each task.

**Model Pretraining Strategy**  During pretraining, we apply teacher forcing and compute the cross-entropy loss between the predicted and target token. As the typical length of the target prediction varies per task, we ensure the losses across tasks are comparable in scale by averaging the loss by the target sequence length first and then by the number of samples in the batch.

We employ a 'mixed batches' pretraining scheme where each batch contains examples sampled from any task. Assuming that task $i$ has $n_i$ examples, the probability of sampling an example from task $i$ from all $j$ tasks is $p_i = n_i / \sum_j n_j$. However, as shown in Table 6, the pretraining tasks have a large variance in the number of available examples, which can lead to poor performance on low-resource tasks. Therefore, similar to Raffel et al. (2020), we re-adjust the probability $p_i$ by limiting the maximum number of examples allowed per task. The limit is controlled by a ratio $R$, which defines how many more samples are included in training task $i$ versus the task with the smallest quantity of examples $n_{min}$. Therefore, $\forall j$ tasks, the final probability of sampling an example from task $i$ becomes $\bar{p}_i = \min(n_i, R \times n_{min}) / \sum_j \min(n_j, R \times n_{min})$. In our experiments we set $R = 3$.

## B  Implementation Details

**Pretraining Setup**  We pretrain our model for 100k steps using a batch size of 2048 and the AdamW optimizer with weight decay of 0.01. We apply a linear learning rate schedule with warm-up for 10K steps and a maximum learning rate of 1e-4. The pretrained model was trained on 8 NVIDIA Tesla V100 GPUs.

**Vision Encoder**  For the Alexa Arena, we fine-tune the VinVL pretrained checkpoint for 300K steps with a batch size of 4. We set the base learning rate to $10^{-4}$ and weight decay to $10^{-5}$ with an SGD optimizer, decaying the learning rate by 0.1 after steps 55K and 75K steps. During training, we use the default preprocessing and image transformations (Han et al., 2021). The model is trained on 4 RTX 2080 Ti GPUs.

**Fine-tuning EMMA on VL downstream tasks**  For all tasks, we fine-tune the pretrained model using LM loss for up to 20 epochs.

**Fine-tuning EMMA on Alexa Arena**  A key feature of EMMA is the ability to predict a 'negative' output for the contextual routing and the visual grounding task. For example, the agent can output `<act><no match>` apple or `no apple` when trying to pick up or find an apple. Since the DTC data do not include these types of outputs, we use visual and CDF augmentations to simulate these instances. In particular, with a probability 50% we convert a 'positive' instance (an instance that the synthetic instruction matches with the visual scene), into a 'negative' one by selecting an image from the train set where the target object is missing.

For the modular and unified models, we fine-tune our pretrained model for 10K steps, using cross-entropy loss and teacher-forcing. We use a batch size of 256 and the AdamW optimizer with learning rate $10^{-4}$, weight decay 0.01, a linear learning rate schedule with 1K warmup steps, and 0.1 label smoothing. We shuffle the identities of the visual tokens for each frame. The model is trained on a single RTX 2080 Ti GPUs.

## C  DTC Benchmark

The DTC benchmark contains 2661 missions in training and 383 in validation. Each mission is annotated by three separate annotators. Each human annotation corresponds to a single episode, meaning that there are 7983 training episodes and 1149 validation episodes.

Primitive navigation actions include `MoveForward`, `MoveBackward`, `RotateLeft`, and `RotateRight`. To collect panoramic views from its position, the agent can perform the `LookAround`

| Dataset | # Images | # Samples | Tasks |
|---|---|---|---|
| VQA-v2 (Goyal et al., 2017) | 83K | 443K | VQA |
| GQA (Hudson and Manning, 2019) | 86K | 987K | VQA |
| COCO Captioning (Lin et al., 2014) | 118K | 592K | MLM, ITM, Captioning |
| Conceptual Captions (Sharma et al., 2018) | 3M | 3M | MLM, ITM, Captioning |
| Visual Genome (Krishna et al., 2017) | 108K | 5.4M | MLM, Dense Captioning, Visual Grounding, Relationship Detection |
| Total | 3.3M | 10.4M | |

Table 6: Dataset statistics for pretraining.

| Task | Example input | Example target output |
|---|---|---|
| MLM | Denoise: Fridge `<MASK>` is open | Fridge door is open |
| ITM | Assess the statement: Fridge door is open | True |
| Captioning | Describe the image | Food inside a refrigerator with its door open |
| Dense Captioning | Describe object `<visual_token_5>` | Silver fridge |
| Visual Grounding | Locate the milk carton | `<visual_token_3>` |
| VQA | What color are the cabinets? | White |
| Relationship Detection | Explain how `<visual_token_3>` relates to `<visual_token_5>` | Milk inside of fridge |

Table 7: Example input and output formats used for the pretrained tasks.

action. The agent also performs manipulation actions on each object which include `PickUp`, `Place`, `Open`, `Close`, `Toggle`, `Fill`, `Clean`, `Pour`, `Break`, and `Scan`. The set of supported actions for each object are determined based on its affordances. In total, there are 14 affordances: `pickupable`, `openable`, `breakable`, `receptacle`, `toggleable`, `powerable`, `dirtyable`, `heatable`, `eatable`, `chillable`, `fillable`, `cookable`, `decor`, and `infectable`.

## D   Search Routine

We focused on the development of a dialogue-enabled agent within an embodied environment. Our design aimed at creating an agent that is as autonomous as possible without relying on external components to support it. Nevertheless, the search routine is an external component that our agent depends upon. Learning how to search for an object is a challenging problem that would require an excessive amount of demonstrations.

The search pipeline is triggered when the output of the contextual routing task requires the agent to search for an object or to interact with an object that is not in the agent's view. The agent searches the current room by iterating through selected viewpoints – including the agent's original position. At each viewpoint, we collect a panoramic view by rotating left by 90 degrees three times.

Since each room has a maximum of eight viewpoints, the search routine could amount to a large number of redundant steps. To mitigate this, the agent selects a subset of viewpoints. We assume that the original agent position and each viewpoint can cover an area of a fixed radius. This way, we create a graph where each candidate position is a node and add edges between nodes whose areas overlap. As a result, we turn the viewpoint selection into the Maximum Vertex Coverage problem and apply a greedy algorithm. We empirically set the radius to four, which leads to selecting up to two viewpoints depending on the room size. After preparing the search plan, the agent starts executing it step-by-step. We use the visual grounding task to localize a referenced object within each new frame. The agent continues to execute the search plan until the object is found or the plan is exhausted.

To minimize the number of search steps, we also keep track of the objects observed at each position. If any of the object labels from the vision model is not present in the memory, we create a new entry that stores the object label, its bounding box area, and the closest viewpoint. Since we are not using depth estimation, we used the area as an approximation of distance. The memory is queried at the beginning of the search routine. If the target object is in memory for the current room, the agent will consider the retrieved viewpoint as the starting position for the search.

A limitation of our setup is that our routine does not keep track of the spatial positions of an object. Some objects may be inside containers like a fridge,

| Task | Prompts |
|---|---|
| MLM | "Denoise: {caption}" |
| | "Denoise the statement: {caption}" |
| | "Denoise the description: {caption}" |
| | "Reconstruct: {caption}" |
| | "Reconstruct the description: {caption}" |
| | "Reconstruct the statement: {caption}" |
| ITM | "Assess the statement: {statement}" |
| | "Assess the description: {statement}" |
| | "Evaluate the statement: {statement}" |
| | "Evaluate the description: {statement}" |
| Captioning | "Caption this" |
| | "Caption the image" |
| | "Caption this image" |
| | "Describe this" |
| | "Describe the image" |
| | "Describe this image" |
| Dense Captioning | "Caption {region}" |
| | "Caption object {region}" |
| | "Describe {region}" |
| | "Describe object {region}" |
| Visual Grounding | "Find the object: {caption}" |
| | "Locate the object: {caption}" |
| | "Pick the object: {caption}" |
| | "Select the object: {caption}" |
| VQA | "Answer: {question}" |
| | "Answer the question: {question}" |
| | "What is the answer to: {question}" |
| | "What is the answer to the question: {question}" |
| Relation Detection | "Explain the relationship between: {subject} and {object}" |
| | "Explain how {subject} relates to {object}" |
| | "Describe the relationship between {subject} and {object}" |
| | "Describe how {subject} relates to {object}" |
| Action Execution | "Act according to the instruction: {instruction}" |
| | "Execute the instruction: {instruction}" |
| | "Follow the instruction: {instruction}" |
| Countextual Routing | "Predict the system act: instruction" |

Table 8: Task prompts used for each pretraining and downstream task.

or a cabinet. If the search routine is triggered after the object has been observed inside a container then the agent would need first to interact with the container before retrieving the object. A more sophisticated semantic search policy (Blukis et al., 2018a,b, 2022; Chaplot et al., 2020; Min et al., 2021) would likely translate to better performance.

## E  Data Augmentations

**Visual Augmentations** We leveraged the images used to fine-tune the object detector to create synthetic instances. Figure 6 shows a comparison between the distribution of objects from the DTC and our synthetic dataset. Our synthetic dataset includes objects that are not used within the DTC benchmark. To prevent overpopulation of the synthetic dataset with objects that occur frequently (tables and desks), we set a maximum number of objects for each action. Table 9 shows the maxi-

| | Train | Validation |
|---|---|---|
| Break | 750 | 400 |
| Clean | 400 | 200 |
| Close | 750 | 400 |
| Fill | 750 | 400 |
| Goto | 750 | 400 |
| Open | 750 | 400 |
| Pickup | 750 | 400 |
| Place | 750 | 400 |
| Pour | 750 | 400 |
| Scan | 400 | 200 |
| Search | 750 | 500 |
| Toggle | 750 | 400 |

Table 9: Maximum number per object for each action. Search instances are used for multimodal grounding which is not supported by the DTC benchmark.

mum number of each object used for every action used for train and validation.

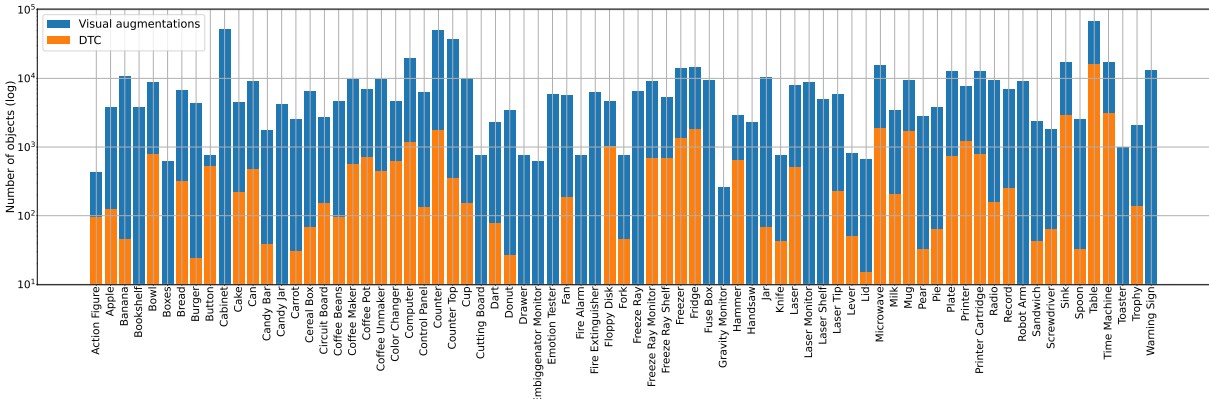

Figure 6: Distribution of objects used in our visual augmentations and the DTC benchmark.

Each image contains ground truth information regarding the position of the agent, the position of the objects as well as their current states. We used the ground truth metadata to ensure the credibility of the synthetic instances. First, we used the distance between the agent and the object to be less than the minimum interaction distance defined by the Alexa Arena. For search instances, we omit this requirement since we want the agent to be able to find any object that is visible from the current view but arbitrarily far away from its position. Second, we used the states of the objects to determine if an action can be executed on an object. For example, the `close` action on a fridge cannot be performed if the fridge is already closed. The `goto` and `scan` actions are executable regardless of the state of the object. In these cases, we ignore the preconditions regarding the state of the object.

**CDF Augmentations**   The objective of the CDF augmentations is to help the agent learn frame and visual token associations. The DTC benchmark includes long trajectories but from our preliminary experiments, we found that the model was not learning these associations, particularly as the number of steps for a single instruction increased.

In order to create CDF augmentations we deployed one of the earliest versions of EMMA in the Alexa Arena. We created missions similar to those in the DTC benchmark and manually wrote low-level instructions so that the model could complete the missions with one or few-shot interactions without having to perform long trajectories. When creating a mission we randomly sampled objects equipped with the affordances that we are interested in for the specific mission. To ensure variety within the missions, we randomized the layout of the room, the position and the color of objects in

the layout, as well as the position of the agent at the start of the mission.

We collected successful mini-episodes and used them to train the model. Note that these mini-episodes were hard for the initial version of EMMA and were only feasible due to external guidance, such as the search routine.

**Paraphrazable Instructions**   To create instructions for both visual and CDF augmentations we used ChatGPT as a starting point. Examples of interactions with the ChatGPT are shown in Figure 7. To create the high-level instruction for each CDF trajectory, we merged the low-level instructions and asked ChatGPT to provide a high-level one. We did not use any sophisticated prompt engineering techniques. We only provided introductory sentences along with a few examples of semantically similar instructions and asked the model to provide paraphrases. Some of the generated paraphrases do not match the target domain. For example, to clean the bowl in the sink, the agent does not require soap. We manually filtered these out before compiling our set of instructions.

## F   Experiments

| Object category | Baseline | Ours |
|---|---|---|
| Small (0–1.3k) | 37.63 | **51.90** |
| Medium (1.3k–9.2k) | 60.41 | **89.60** |
| Large (9.2k–90k) | 64.72 | **91.90** |
| All | 46.03 | **56.70** |

Table 10: Object detection results for small, medium and large objects. The allowed area of an object in each category is shown in parentheses.

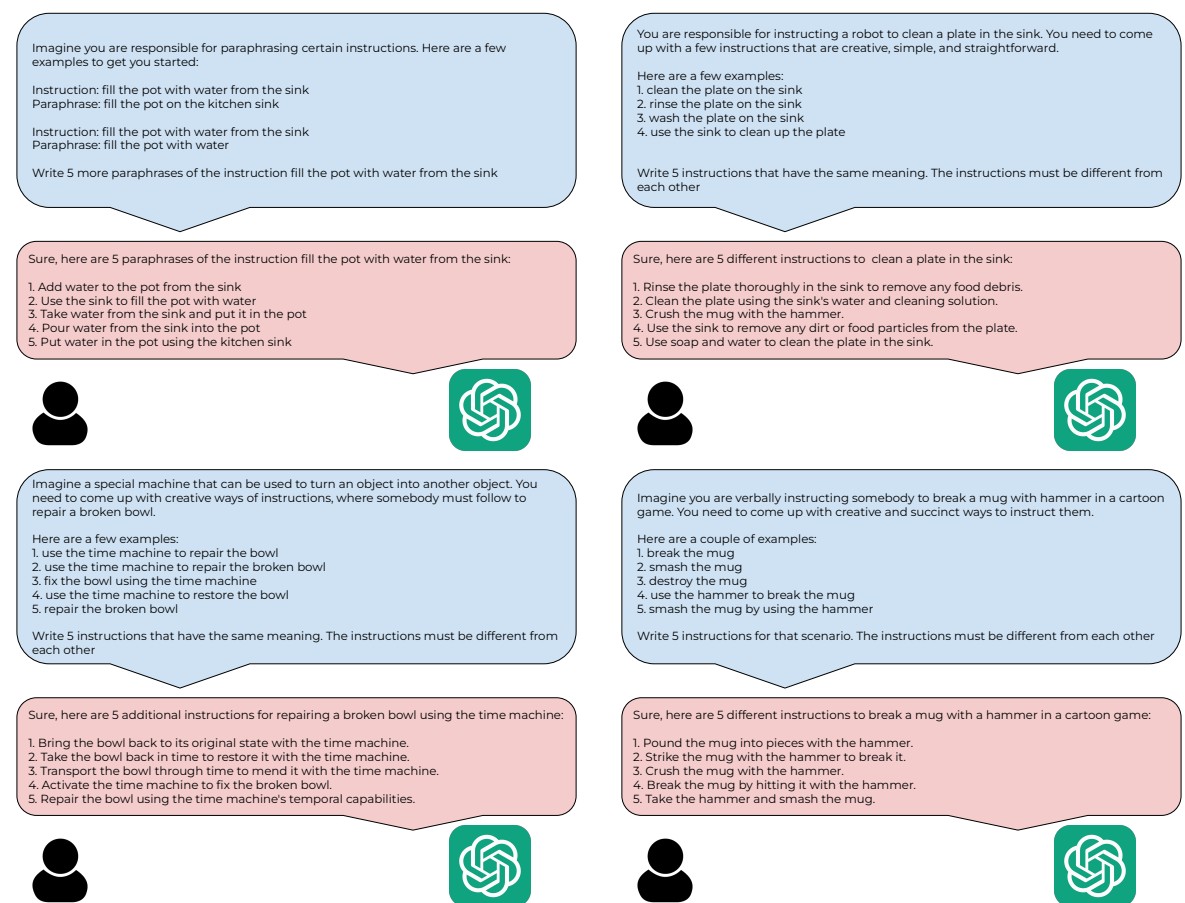

Figure 7: Examples of using ChatGPT as a starting point for paraphrasing goals and subgoals within the Alexa Arena.

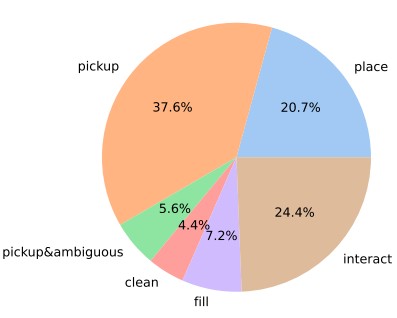

Figure 8: Distribution of CDF augmentations during training.

| Model | Accuracy | F1 |
|---|---|---|
| EMMA-modular | 0.96 | 83.33 |
| EMMA-unified | 0.97 | **84.67** |

Table 11: Offline evaluation of the CR task.

**Task-specific Prefixes** Previous approaches have shown that variations in the text prompt can affect the results on downstream performance (Cho et al., 2021; Gao et al., 2021; Radford et al., 2021). We initially experimented with two types of prefixes: task-specific special tokens and natural language prompts. Task-specific special tokens are single-word descriptors (Cho et al., 2021), while natural language prompts are longer, varied descriptions similar to the approach proposed by Sanh et al. (2022). For example, the image captioning task is denoted either by the single token [Cap] or by prompts such as 'Describe this' or 'Caption the image'. Before pretraining our EMMA-base we explored this design choice using an EMMA-small variant following BART architecture (Lewis et al., 2020). In particular, we observed the validation loss of the model with tag and text prefixes after pretraining for 75K steps. The model using textual prefixes was slightly outperforming the model using tags (0.818 vs 0.824) which aligns with previous findings (Cho et al., 2021). For this reason,

we chose to use natural language prompts but also they allow for a more flexible interface with the model which is especially useful if the downstream task involves language variety and complexity.

**Data Ablations** When training solely on the DTC data, the baseline model outperforms EMMA. To verify that our approach scales better in terms of the data, we also experiment with applying our augmentations to the existing baseline. We train the baseline model using the same set of hyperparameters as in (Gao et al., 2023) on the DTC data and vision augmentations. Then, we compare it against our model trained on the same data (see red curve in Figure 4). The baseline model achieves a 34.42% success rate, a small performance boost as opposed to training with only the DTC data, while our model benefits substantially from these augmentations achieving a success rate of 34.72%.

**Model Scaling** We also experiment with a small variant of our model (3 encoder layers, 3 decoder layers, 8 attention heads, 368 hidden size, 512 feed-forward size) with a total of 20M parameters and the base model trained from scratch on the entire dataset. The performance of the small variant is 32.11%, while the base model trained from scratch performed poorly with a success rate of 5.2%. Our results indicate that scaling the model's size up provides a substantial performance increase and highlights the necessity of pretraining.

**Object Detector Comparison** We validate the performance of our vision encoder by comparing it against the baseline (Gao et al., 2023). We evaluate our object detection model using the standard COCO evaluation metric, the Mean Average Precision (mAP), calculated by averaging the precision at IoU thresholds ranging from 0.5 to 0.95 in steps of 0.05. Similar to Gao et al. (2023), we set the maximum detection proposals to 100 for evaluation. Table 10 compares our vision encoder with the baseline across different object sizes as defined in (Gao et al., 2023). Our model achieves strong performance with approximately 40% relative improvement for all object sizes. We note, however, that our vision encoder is not directly comparable to the baseline since the baseline model is trained on fewer classes (86 vs 133) and also performs segmentation instead of object detection.

**Offline Contextual Routing Performance** Section 7 concludes that CR is one of the main sources of errors because as the first step in completing an instruction, an error will have cascading effects. We evaluate offline the model performance on the CR task using the validation set of DTC. Table 11 shows the accuracy and macro-average F1 score for the model predictions of the CR-specific tokens. Both the modular and unified models achieve high scores. However, assuming that action execution and search are performed without errors, the probability of mission success for a mission with 5 instructions is $(1 - 0.03)^5 = 0.85$.

# G Examples of trajectories

In Section 7 we highlighted the main source of errors; the output of the contextual routing task and often the lack of temporal understanding when our model acts on the environment. We observed that when performing the contextual routing task the model can be confused when there are multiple object candidates in the scene that could be suitable candidates for an instruction. When performing multiple actions over the same object, the model may use a token to reference the object in a previous timestep as opposed to the current one. We show two examples of such errors in Figure 9.

# H Visual Domain Transfer

We create a synthetic dataset based on scene graphs from the validation set of GQA (Hudson and Manning, 2019), which provides a cleaned version of Visual Genome scene graphs (Krishna et al., 2017). This toy dataset is created to estimate the model's ability to transfer the downstream task of action execution to the pretraining visual domain, and quantify the impact of using the base or fine-tune object decoder. We generate synthetic instructions by populating templates for the action types: *go to*, *pick up*, *place*, *open*, *close*. The templates are in the form of '{action verb synonym} the {optional attribute} {target object}', e.g. 'head towards the wooden table'. We generate *go to* instructions for all objects, and use the affordances from Alexa Arena to determine if the object class is compatible with the remaining action types. We want to avoid cluttered images and ambiguous instructions. Therefore, we keep images with up to 10 objects and skip objects whenever multiple objects of that class appear in the image. We create a total of 57740 instructions.

We also investigate the ability of the fine-tuned model to perform the pretraining tasks in the down-

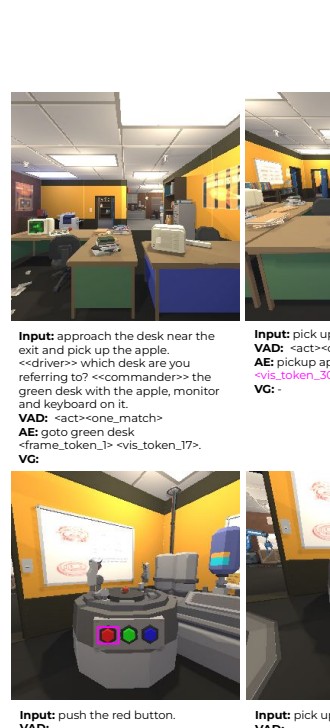
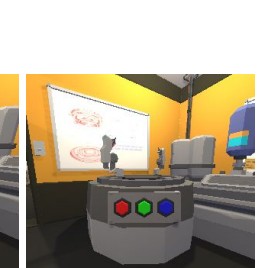
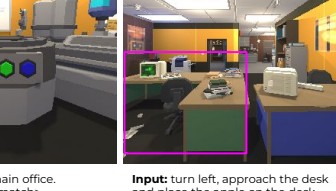
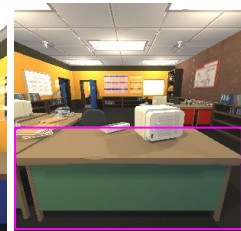

**Input:** approach the desk near the exit and pick up the apple. <<driver>> which desk are you referring to? <<commander>> the green desk with the apple, monitor and keyboard on it.
**VAD:** <act><one_match>
**AE:** goto green desk <frame_token_1> <vis_token_17>.
**VG:**

**Input:** pick up the apple
**VAD:** <act><one_match>
**AE:** pickup apple <frame_token_2> <vis_token_30> <stop>.
**VG:** -

**Input:** go to the quantum lab.
**VAD:** <act><one_match>
**AE:** goto lab2 <stop>.
**VG:** -

**Input:** approach the color changer station and put the apple on it.
**VAD:** <act><one_match>
**AE:** goto color changer <frame_token_1> <vis_token_4>.
**VG:** -

**Input:** approach the color changer station and put the apple on it.
**VAD:** <act><one_match>
**AE:** place color changer <frame_token_2> <vis_token_3> <stop>.
**VG:** -

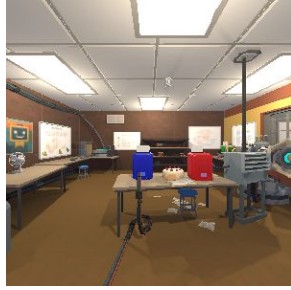
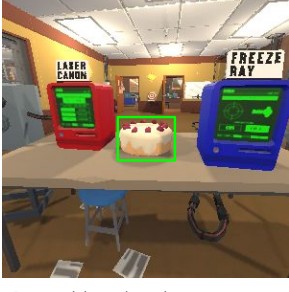
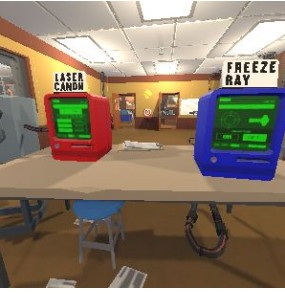
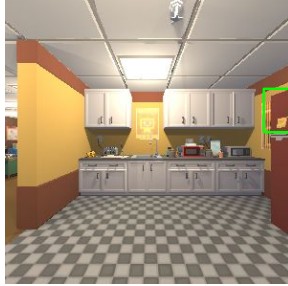

**Input:** push the red button.
**VAD:** -
**AE:** toggle red button <frame_token_1> <vis_token_6> <stop>.
**VG:** -

**Input:** pick up the red apple.
**VAD:** -
**AE:** pickup apple <frame_token_1> <vis_token_10> <stop>.
**VG:** -

**Input:** go to the main office.
**VAD:** <act><one_match>
**AE:** goto mainoffice <stop>.
**VG:** -

**Input:** turn left, approach the desk and place the apple on the desk
**VAD:** <act><one_match>
**AE:** goto green desk <frame_token_1> <vis_token_17>.
**VG:** -

**Input:** turn left, approach the desk and place the apple on the desk
**VAD:** <act><one_match>
**AE:** place green desk <frame_token_2> <vis_token_4> <stop>.
**VG:** -

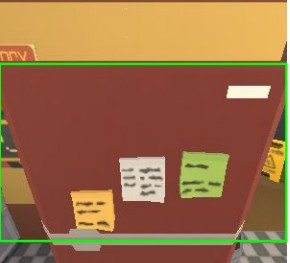
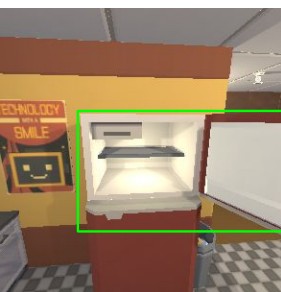
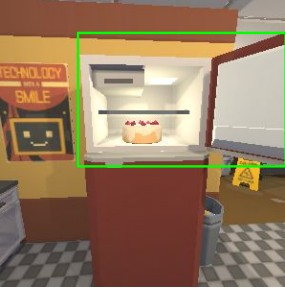
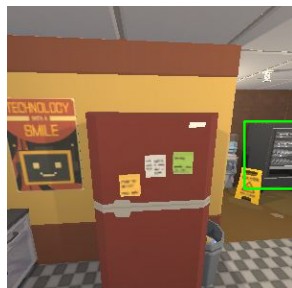

**Input:** find cake
**VAD:** <search><one_match>
**AE:** -
**VG:** <frame_token_1> <vis_token_1>

**Input:** pick up the cake
**VAD:** <act><one_match>
**AE:** pickup cake <frame_token_1> <vis_token_1> <stop>.
**VG:** -

**Input:** go to the break room
**VAD:** <act><one_match>
**AE:** goto breakroom <stop>.
**VG:** -

**Input:** find upper fridge
**VAD:** <search><one_match>
**AE:** -
**VG:** <frame_token_1> <vis_token_8>

**Input:** freeze cake in upper fridge.
**VAD:** <act><one_match>
**AE:** open freezer <frame_token_1> <vis_token_1>.
**VG:** -

**Input:** freeze cake in upper fridge.
**VAD:** -
**AE:** place freezer <frame_token_2> <vis_token_6>.
**VG:** -

**Input:** freeze cake in upper fridge.
**VAD:** -
**AE:** close freezer <frame_token_3> <vis_token_8>.
**VG:** -

**Input:** freeze cake in upper fridge.
**VAD:** -
**AE:** open freezer <frame_token_4> <vis_token_8>.
**VG:** -

Figure 9: Example of two unsuccessful trajectories. The objective of the top mission is to use the color changer on the apple and deliver it to the red desk. The instruction at the second to last timestep requires the agent to place the apple on the red desk. In the bottom mission, the objective is to freeze the cake in the freezer and deliver it to the desk. The agent at the last time step tried to open the vending machine (visual token 8) which is the token to reference the freezer in the previous timestep.

stream visual domain through qualitative examples. This requires the model not to forget the pretraining tasks and to demonstrate generalization to the visual domain of the downstream tasks. To improve model outputs, we prohibit the generation of added special tokens (frame, visual, stop and CR tokens). The examples in Figure 10 show that the fine-tuned model retrains VQA, dense captioning and relationship detection capabilities and transfers them to the simulated environment domain. Figure 11 shows some additional examples of the action prediction task on real images.

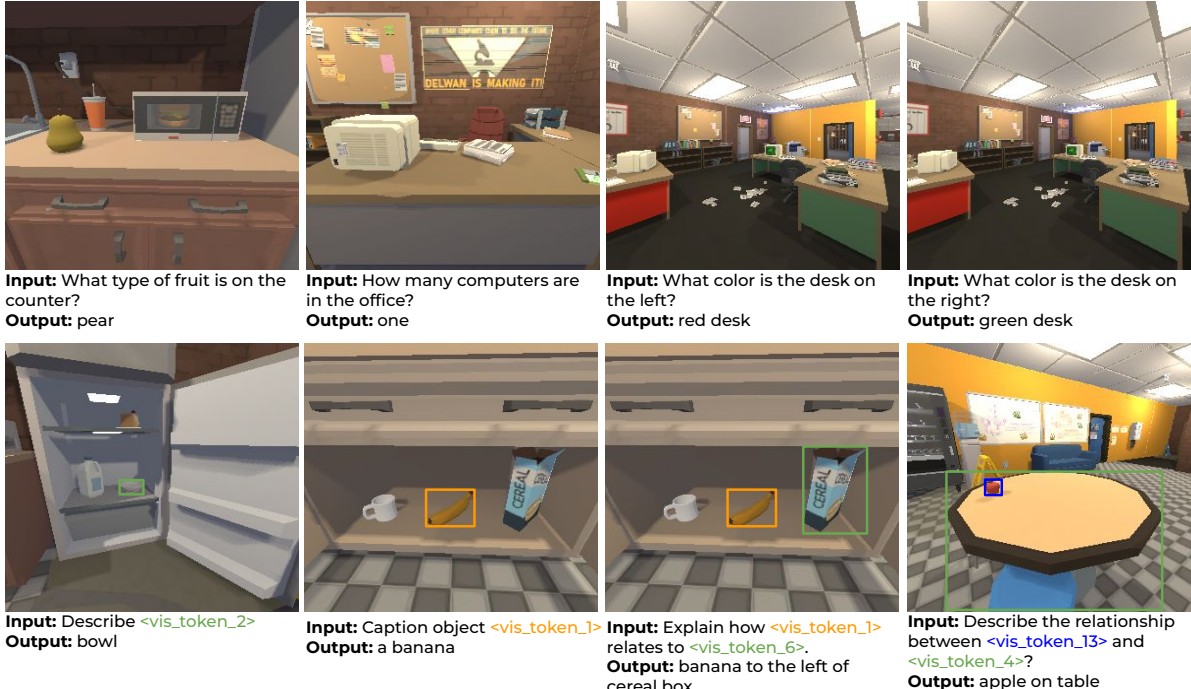

Figure 10: Examples of transfer of pretraining tasks to the downstream domain after finetuning the pretrained model.

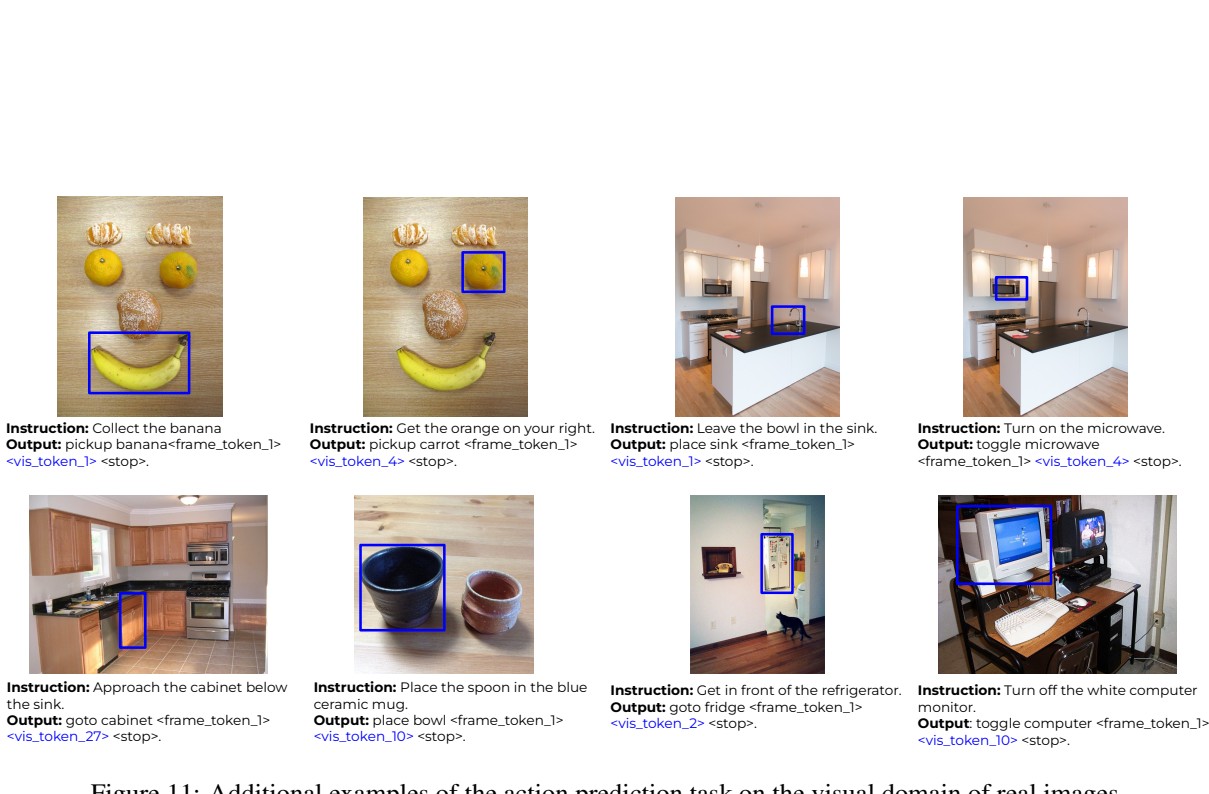

Figure 11: Additional examples of the action prediction task on the visual domain of real images.