# OpenReview forum: "Multitask Multimodal Prompted Training for Interactive Embodied Task Completion"
_EMNLP/2023/Conference — EMNLP 2023 Main_

### Official Review · Reviewer_eEZM · 2023-07-29

**Soundness:** 4

**Excitement:**

4: Strong: This paper deepens the understanding of some phenomenon or lowers the barriers to an existing research direction.

**Paper Topic And Main Contributions:**

The main topic of the paper is the development of an Embodied MultiModal Agent (EMMA) to address the challenges of interactive and embodied tasks in Vision & Language (VL) models. The key contributions include:

1. Proposing a single, unified encoder-decoder model that reasons over images and trajectories, and casts action prediction as multimodal text generation.
2. Achieving state-of-the-art performance (36.81% success rate) on the Dialog-guided Task Completion (DTC) benchmark in the Alexa Arena.

**Reasons To Accept:**

- The motivation and the identification of current challenges on leveraging VL models for interactive and embodied tasks are accurate and inspiring.
- The proposed EMMA model has simple yet effective architecture with the key idea of unifying everything into text generation.
- The paper is well-written and easy-to-follow.

**Reasons To Reject:**

1. Although it’s probably a concurrent work, it would be great to have some comparison between EMMA and PaLM-E (https://palm-e.github.io/). The main idea seems to be very related.


**Reproducibility:**

5: Could easily reproduce the results.

**Reviewer Confidence:**

4: Quite sure. I tried to check the important points carefully. It's unlikely, though conceivable, that I missed something that should affect my ratings.

---

> ### Author Rebuttal · Authors · 2023-08-28
>
> We thank reviewer eEZM and we are happy that they found the motivation for our approach inspiring and our unified model effective.
>
> PaLM-E is indeed concurrent work exploring similar themes that we have cited in our related work section. However, the model is not publicly available and additionally it poses significant computational cost due to its scale. We have opted for a more lightweight model whose training can be reproduced using publicly available data and using a modest computational budget (8 x V100 for pretraining, 1 GPU with 11GB for finetuning). In order to facilitate reproducibility, we will release our experimental framework together with model checkpoints, training data and training configurations. We will add these important considerations in the final version of the paper.

---

### Official Review · Reviewer_GWnB · 2023-08-04

**Soundness:** 4

**Excitement:**

4: Strong: This paper deepens the understanding of some phenomenon or lowers the barriers to an existing research direction.

**Paper Topic And Main Contributions:**

The authors propose a novel embodied agent model for the Dialog-Guided Task Completion benchmark in the Alex Arena. The agent is based on a single vision-language backbone that is used to complete all subtasks such as visual grounding and action execution. To this end the authors pre-train their own vision-language model from scratch and benchmark it against existing VL models. To train their agent the authors further introduce two data augmentation methods which significantly boost performance and enable the model to perform the task successfully. Finally, they evaluate their approach extensively both quantitatively and qualitatively and compare it to baselines and the Alexa Arena leaderboard.

**Questions For The Authors:**

Question A: What are the task-specific prompts used for the subtasks?

Question B: I'm curious how the visual grounding task works when the object is not found immediately. When you invoke the search algorithm described in Appendix D do you simply select two out of four available viewpoints per room that maximize vertex coverage in the created viewpoint graph? Does that mean you simply iterate through the four directions for each of of the two viewpoints for a maximum of eight possible views until the object is found? In that case does the agent simply navigate through search and object memory without any of the primitive navigation actions being directly invoked by EMMA?

Question C: You say that "when interacting with a fridge across multiple frames, the agent sometimes predicted visual tokens that matched the fridge in a previous rather than the current frame." (lines 317-320). Do you find that this behavior, and therefore grounding to the current frame, has improved after data augmentation?

Question D: Have you experimented with integrating differently sized pre-trained models (i.e. a smaller model or one that was trained on fewer samples) or larger existing pre-trained models such as OFA-base into your agent? It would be interesting to see how the agent's task performance scales with the performance of the pre-trained model on vision-language tasks.

Question E: How does your model compare to the baseline VL model by Gao et al. in terms of model size? How much of the improvement over the baseline can be attributed to using a better VL model?

Question F: In Figure 4, is this ablation with or without clarification questions? If it with clarifications, why is performance without augmentations so much lower than the VL baseline? I would expect your proposed augmentations to yield benefits for the VL baseline as well so I am curious whether you have done any experiments applying your augmentations to the existing baseline and how your model compares.

Question G: This is more of a suggestion but I think the results from Figure 10 and the simulation to real-world transfer that you mention are quite interesting and deserve to be explored more.

**Reasons To Accept:**

The paper is well written and deals with the important topic of grounding language in vision and actions through a proposed embodied agent model for the DTC benchmark.

The proposed method achieves promising results and in particular the data augmentation methods seem to yield significant improvements which make this sort of approach possible.

The evlauation is thorough and contains a separate performance evaluation for success rates by category and question type as well as augmentation ablations and qualitative error analysis.

I also found the results when transfering the model to real images quite interesting.

**Reasons To Reject:**

Although the paper is mostly well written sometimes details can be a bit unclear (see questions) or require re-reading or checking the appendix. For example a clearer definition of model inputs and outputs would be helpful.

In particular I would be interested in seeing the particular task-specific prompts used for the CR, AE and VG subtasks.

Although evaluation results are promising the novelty is somewhat limited: benefits over the existing VL baseline (Gao et al., 2023) from the original DTC paper are small and it is unclear if the comparison between EMMA and this VL baseline are fair in terms of parameter counts and computational costs. It seems simply using a larger model trained on more data would likely yield better results so some more discussion of this aspect of the evaluation would be helpful.

Update after rebuttal:

The authors have sufficiently addressed unclear details, answered all of my remaining questions and stated they will also add the full prompts to the final papers. Given that most of my concerns have been addressed, I have decided to raise the soundness score of my review.

**Reproducibility:**

3: Could reproduce the results with some difficulty. The settings of parameters are underspecified or subjectively determined; the training/evaluation data are not widely available.

**Reviewer Confidence:**

3: Pretty sure, but there's a chance I missed something. Although I have a good feel for this area in general, I did not carefully check the paper's details, e.g., the math, experimental design, or novelty.

**Typos Grammar Style And Presentation Improvements:**

While I understand the amount of content that can be fit into the main paper is limited some very important information (like the search routine which describes how the agent navigates and therefore seems to be very relevant) is only contained in the Appendix. It would be helpful if the paper was re-structured a little and with references to the Appendix where appropriate.

---

> ### Author Rebuttal · Authors · 2023-08-28
>
> We thank the reviewer GWnB for their insightful feedback! We are very pleased that they find the topic of grounding language in vision and actions an important research problem. We are glad that they found our evaluation solid as well as the transferring of the model to real images appealing. We address the questions below:
>
> Re: clearer definition of model inputs and outputs
>
> We agree that some details require checking the appendix. The task-specific prompts for CR (Contextual Routing) and AE (Action Execution) are not presented in the paper. An example of the VG (Visual Grounding) input-output format is shown in Table 7 for the pretraining though the exact same format used during the downstream embodied task. We provide some examples for the CR and the AE tasks in Question A and we will add the full prompts in the final version of the paper. Finally, Table 7 shows examples of input-output for each pretraining task.
>
> Additionally, we would like to point to Section 4.2 that provides a more descriptive of the model inputs and outputs for each downstream task.
>
> CR: (line 268-280 & Figure 3) the model accepts the current instruction and observation and outputs a sequence where the first special token determines whether the instruction refers to an action or to search for an object, while the second special token indicates if the object can be matched in the current view. Finally, for manipulation actions the model also generates the name of the object.
>
> AE: (lines 284-295 & Figure 1) the model accepts the observations up to the current time step and the instruction prompt, which includes the instruction and if applicable the inventory object and clarification. The output of the model consists of the action type followed by the name of the object as well as the special frame and visual tokens used to reference the object in the trajectory. Each predicted action results in a new observation which we combine with the previous context to make a new prediction
>
> VG: (lines 298-305 & Figure 2) we invoke the search routine and for each view we use the pretraining VG prompt along with the current view and the instruction. If the object is present in the view the model outputs the special tokens that match it otherwise it outputs ‘no OBJECT NAME’.
>
> We hope that this is sufficient to understand the behavior of our model and we are happy to restructure any specific sections that might seem confusing or include additional details that you believe are missing. The final version of the paper will be proofread to clarify any unclear details, which we also address in the questions below.
>
> Re: Novelty & benefits over VL baseline
>
> Please see the response to the questions below.
>
> [Question A] We used the following prompts for each of the pretraining tasks. We will add a complete list of prompt templates in the appendix.
>
> * MLM: "Reconstruct the description: {caption}"
> * ITM: "Evaluate the description: {statement}"
> * Captioning: "Caption this image"
> * Dense Captioning: "Describe object {region}"
> * Relationship Detection: "Explain the relationship between: {subject} and {object}"
> * VQA: "Answer the question: {question}"
> * Visual Grounding: "Locate the object: {caption}"
> * Contextual Routing: "Inventory: {inventory}. Predict the system act: <<commander>> {instruction}"
> * Action Execution: "Inventory: {inventory}. Act according to the instruction: <<commander>> {instruction} <<driver>> {question} <<commander>> {answer}"
>
> [Question B] That is correct. Following the DTC API definition, the agent explores the rooms by going to different viewpoints and approaches objects by selecting their bounding box. This is in contrast with other benchmarks such as ALFRED (Shidhar et al., 2020) that do not have such option.
>
> [Question C] Yes, we find that the CDF augmentations are required for the model to learn frame-visual token associations. Note that our pretraining consists of image-caption pairs which allows the model to learn to ground referring expressions to objects in the image. The skill of learning frame-visual token associations is obtained during fine-tuning only. During the DTC task, the agent has to take into account the temporal order of objects, indicated by the frame tokens. We will consider for future work how to define pretraining tasks that favor learning this ability as well.
>
> [Question D] Yes, that is an exciting experiment that deserves investigation.
>
> During the rebuttal period, we also experimented with a small variant of our model (3 encoder layers, 3 decoder layers, 8 attention heads, 368 hidden size, 512 feedforward size) with a total of 20M parameters. The performance of the small variant is 32.11%. We also experiment with the base-model trained from scratch on the entire dataset (DTC + augmentations). The base model trained from scratch performed poorly with a success rate of 5.2%, meaning that pretraining such agents is essential. Unfortunately in the limited amount of time and with limited sources, we were unable to scale up the size of the model. We will add these results to the final version of the paper.
>
> Adding OFA-base could potentially yield higher performance. We note however, that OFA-base employs patch representations which may not be as beneficial as object-centric representations for embodied tasks (Driess et al., 2023; Jiang et al., 2023). We discuss this issue in lines 220-233. We believe that this is an important research question that deserves a more controlled comparison between patch-embeddings and object-centric representations in embodied tasks. However, we leave this additional experiment for future work considering that it would require additional computational resources because OFA-base is bigger than EMMA (69M more parameters) and requires longer input sequences (due to patch embeddings). Additionally, adapting OFA for embodied tasks requires substantial changes that we are not able to complete in the short time available for the rebuttal.
>
> [Question E] The baseline VL model is a specialized / task-specific architecture with 152M trainable parameters while EMMA has only 113M. Each frame and instruction are encoded with CLIP-ResNet-50 text and image encoders followed by a joint 3-layer transformer encoder. Then representations for each time step are passed to a two-layer LSTM and three separate classification heads for action prediction, object class prediction and end of sequence prediction. The object mask is predicted by additional deconvolutional layers.
>
> Our model follows a simpler unified architecture that jointly predicts actions and bounding boxes (specified by the frame and visual tokens). This approach is appealing because it’s easier to scale to bigger and more diverse datasets involving multiple tasks.
>
> [Question F] Interesting question! The ablation is with clarification questions. We were also wondering why the performance of our model without augmentations is worse than the baseline. We hypothesize that this is due to treating the problem as a classification as opposed to generation. We also note that, on top of predicting actions and object masks, we have the contextual routing (CR) as an additional task.
>
> We also experiment with applying our augmentations to the existing baseline. Unlike the baseline, our approach does not model the “look around” actions which makes the CDF augmentations not useful for the baseline model. We therefore train the baseline model with only the vision augmentations and the DTC data and compare it with our model trained on the same data (see Figure 4, first bullet on the red curve). During training we used the same set of hyperparameters as in (Gao et al., 2023). This model achieves a 34.42% success rate, a small performance boost as opposed to training with only the DTC data (34.20%) and is comparable with EMMA’s success rate of 34.72%. Our model benefits substantially from the augmentations as opposed to the baseline model, which we believe is due to the improvement in the performance of the contextual routing (CR) as this is the only task that is not part of the baseline model.
>
> [Question G] We agree that this is a very promising aspect for future work. We construct a synthetic evaluation set to measure transfer of the downstream action execution tasks to real world images. In addition, we provide qualitative examples of transfer ability of the pretraining tasks (see Figure 5). Unfortunately, creating an evaluation set for this would require human annotations which we will consider in future work. We also conducted a preliminary qualitative evaluation of the VL baseline model and found that this model is unable to adapt to real images making our approach more appealing for sim-to-real adaptation.

---

### Official Review · Reviewer_GKE9 · 2023-08-06

**Soundness:** 3

**Excitement:**

3: Ambivalent: It has merits (e.g., it reports state-of-the-art results, the idea is nice), but there are key weaknesses (e.g., it describes incremental work), and it can significantly benefit from another round of revision. However, I won't object to accepting it if my co-reviewers champion it.

**Paper Topic And Main Contributions:**

This paper studies interactive and embodied tasks in vision-language. They propose a single mutil-task model, that takes images and trajectories as input and casts action prediction as multimodal text generation, which performs on par with similar models on several VL benchmarks.

**Questions For The Authors:**

1. Can you further elaborate the reason to put Table-1 in the main paper? Comparing with very few previous VLMs is not a very useful experiment to show. Do you mean your model is a new and better VLM with its advanced pretraining/sampling method?
2. More implementation details should be added.

**Reasons To Accept:**

1. The main focus of training a mutil-modal agent is an interesting topic.
2. Their method achieves better performance on the Dialog-guided Task Completion benchmark.

**Reasons To Reject:**

1. The model and pretraining introduces few insights or contributions comparing to previous vision-language models and existing work.
2. Some implementation details may be missing and should be improved. e.g., what is the language decoder used for main results?
3. Though the paper claims to focus on interactive embodied tasks, table-1 compares with an incomplete list of some previous vision-language models on some of the standard VL tasks such as image captioning and VQA. Results are not fully explained.

**Reproducibility:**

2: Would be hard pressed to reproduce the results. The contribution depends on data that are simply not available outside the author's institution or consortium; not enough details are provided.

**Reviewer Confidence:**

3: Pretty sure, but there's a chance I missed something. Although I have a good feel for this area in general, I did not carefully check the paper's details, e.g., the math, experimental design, or novelty.

---

> ### Author Rebuttal · Authors · 2023-08-28
>
> We thank reviewer GKE9 for their thoughtful feedback. We are pleased that they find our approach interesting. We report below the answers to the comments and questions they have made.
>
> Re: few novel insights and contributions
>
> While a unified pretraining is not novel (e.g., Cho et al 2021), applying a unified approach to an embodied task offers new insights because it enables the agent to learn a language of actions that addresses two fundamental challenges within embodied AI: 1) grounding language in trajectories of actions and observations, and 2) referential disambiguation.
>
> More importantly, our approach departs from current embodied AI approaches that use an ensemble of independently trained models, or use separate heads to independently predict an action and optionally an object class and mask (Blukis et al.,; 2022, Min et al., 2021; Pashevich et al., 2021; Singh et al., 2021). Our model, on the other hand, generates actions and references to objects in the environment as text tokens enabling better performance, higher flexibility, and increased ability to scale up the training.
>
> Furthermore, pretraining this type of architecture is necessary; an EMMA model trained from scratch achieves 5.22% success rate while our pretraining setup enables our model to achieve 36.81%. This highlights the importance of multitask pretraining, especially for learning the special tokens used to reference objects. Finally, our work shows that, thanks to the learned language of actions, it is possible to achieve strong performance for task completion and the potential to transfer to real images.
>
> We will add a section showing the benefits of our pretraining in the final version of the paper.
>
> Re: missing details
>
> [Language decoder] As specified in Section 4 (lines 188-189), the EMMA model it’s an encoder-decoder model based on the BART-base architecture. As shown in Figure 2, the language decoder is used to generate the outputs for all downstream tasks, including the V&L tasks and DTC.
>
> [Implementation details missing]  As the reviewer GWnB also pointed out, in the final version of the paper we will include additional implementation details, for example the prompts used during pretraining / finetuning.
>
> We believe that our approach is reproducible because: 1) we will release our experimental framework based on PyTorch Lightning and Huggingface Transformers together with model checkpoints and inference scripts, 2) we will release all the data and training configurations required to reproduce our model, 3) Appendix A specifies the sampling strategy used to balance the pretraining tasks and 4) Appendix B reports the hyperparameters for pretraining and finetuning our model.
>
> Given this information, we believe that our setting is reproducible, but we are happy to include any further details suggested by the reviewers.
>
> Re: incomplete comparisons
>
> [Comparison with other VL - Table 1] Our objective was to explore if advances in unified V&L models can successfully transfer to dialog-based task completion in an embodied environment. This introduces the following four constraints:
>
> 1) a unified encoder-decoder architecture where different tasks are defined by text prompts
> 2) the ability to perform visual grounding in order to complete the embodied task
> 3) object-centric representations due to inductive bias for interacting with objects in the environment and limits to input / output sequence lengths
> 4) a lightweight model architecture that can facilitate running the costly evaluation in the simulated 3D environment.
>
> We consider Table 1 as essential because, before evaluating EMMA on embodied AI tasks, it allows us to verify the effectiveness of the pretraining compared to other comparable encoder-decoder models. By comparable we mean models that: 1) are architecturally similar to EMMA (i.e. encoder-decoder models), 2) have comparable size with our EMMA, 3) have visual grounding capabilities (i.e., ability to refer to objects in the scene), and 4) are open-source to ensure future reproducibility. The only available models that fully satisfies the above criteria were VL-BART / VL-T5, which are both outperformed by EMMA on the visual grounding task with the same number of visual grounding instances during pretraining. Additionally, we report UniTab and OFA-base as additional comparisons because they are encoder-decoder models that use patch embeddings. The former is outperformed by EMMA while the latter has a higher parameter count than our model. It is also important to underline that it’s essential to have a lightweight model for Embodied AI task considering that the model has to undergo an expensive inference process when interacting with the environment in real time.
>
> We will incorporate this justification at the beginning of Section 6.1 to clarify this point raised by the reviewer.

---

### Meta-Review · Area_Chair_LrtN · 2023-09-18

**Recommendation:** 4

**Metareview:**

All reviewers viewed this topic as important, and appreciated the strong results in this paper on the DTC benchmark in the Alexa Arena. While reviewers identified that the model and pretraining procedures largely follow past work, none viewed this as a crucial weakness and I agree with reviewer eEZM, and the authors' response to reviewer GKE9 -- that predicting actions and object references as text is an interesting and likely scalable approach. Some clarification concerns identified by one reviewer were also resolved sufficiently in the author response.

---

### Decision · Program_Chairs · 2023-10-07

**Decision:**

Accept-Main

**Comment:**

All reviewers viewed this topic as important, and appreciated the strong results in this paper on the DTC benchmark in the Alexa Arena. While reviewers identified that the model and pretraining procedures largely follow past work, none viewed this as a crucial weakness and I agree with reviewer eEZM, and the authors' response to reviewer GKE9 -- that predicting actions and object references as text is an interesting and likely scalable approach. Some clarification concerns identified by one reviewer were also resolved sufficiently in the author response.